# LRTI-VSR: Learning Long-Range Refocused Temporal Information from short video clips

## Abstract

Video super-resolution (VSR) can achieve better performance compared to single image super-resolution by additionally leveraging temporal information. In particular, the recurrent-based VSR model exploits long-range temporal information during inference and achieves superior detail restoration. However, effectively learning these long-term dependencies within long videos remains a key challenge. To address this, we propose LRTI-VSR, a novel training framework for recurrent VSR that efficiently leverages **L**ong-**R**ange **R**efocused **T**emporal **I**nformation. Our framework includes a tailored training strategy that utilizes temporal propagation features from long video clips while training on shorter video clips. Additionally, we introduce a refocused intra&inter-frame transformer block which allows the VSR model to selectively prioritize useful temporal information through its attention module while further improving inter-frame information utilization in the FFN module. We evaluate LRTI-VSR on both CNN and transformer-based VSR architectures, conducting extensive ablation studies to validate the contribution of each component. Experiments on long-video test sets demonstrate that LRTI-VSR achieves state-of-the-art performance while maintaining training and computational efficiency.

## 1 Introduction

Unlike single image super-resolution (SISR) which relies solely on intra-frame information to estimate missing details, video super-resolution (VSR) additionally leverages temporal information to reconstruct high-resolution frames. This key distinction allows VSR to achieve a more accurate recovery of the current frame by exploiting temporal redundancy within video sequences. Driven by this benefit, the effective use of long-range temporal information has emerged as a critical research focus, playing a pivotal role in the success of modern VSR models.

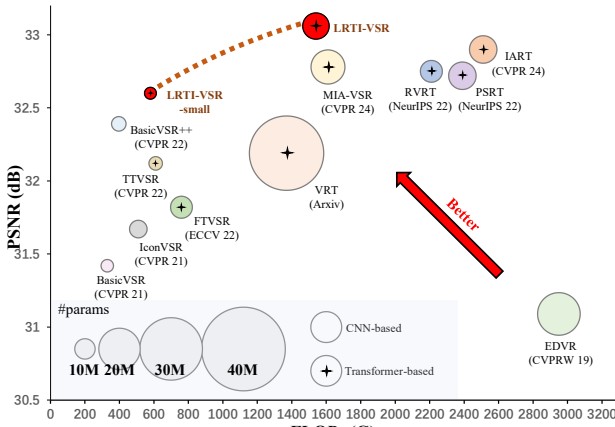

Figure 1: **PSNR(dB) and FLOPs(G) comparison on the REDS4 (Nah et al., 2019) dataset.**

Early-stage research (Li et al., 2020; Wang et al., 2019; Tian et al., 2020; Liang et al., 2022a; Cao et al., 2021) primarily employed a sliding temporal window strategy, utilizing sophisticated alignment modules and advanced network architectures to generate high-resolution (HR) outputs from multiple low-resolution (LR) inputs. As the sliding temporal window limits the utilization of temporal information to a fixed window size, recurrent-based VSR models which propagate hidden states across the whole video begin to occupy a major position in the literature of VSR. In this context, numerous advanced information propagation strategies (Fuoli et al., 2019; Isobe et al., 2020a; Chan et al., 2021; 2022a; Isobe et al., 2022; Shi et al., 2022; Zhou et al., 2024) have been proposed to make more accurate use of temporal information to achieve better VSR performance. Combined with cutting-edge network architectures, the recurrent-based VSR modeles have

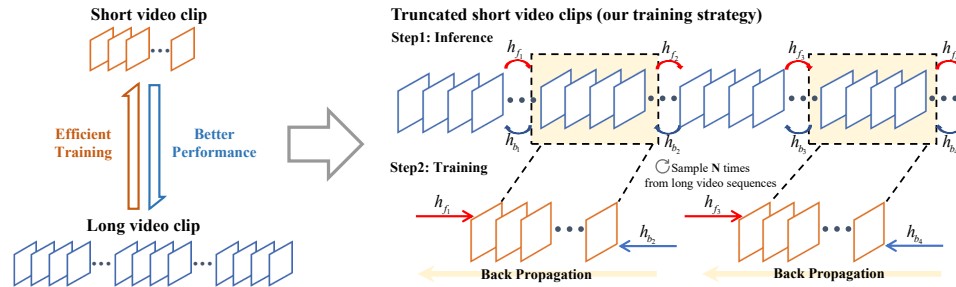

Figure 2: **The pipline of our proposed training strategy.** Our proposed training strategy can utilize accurate temporal dependencies in long video sequences to assist training while using short video clips for high training efficiency. In this figure, $\longrightarrow h_f$ means forward propagation hidden state and $\longleftarrow h_b$ means backward paopagation hidden state in the bidirectional recurrent-based VSR model.

significantly advanced the state-of-the-art in VSR field (Chan et al., 2022a; Liang et al., 2022a; Qiu et al., 2022; Liu et al., 2022; Liang et al., 2022b; Shi et al., 2022; Xu et al., 2023; Zhou et al., 2024).

While recurrent-based VSR models have demonstrated strong capabilities in exploiting long-range temporal information, their training becomes increasingly challenging when longer video clips are used as input due to their sequential dependency constraints. Recent studies (Chan et al., 2021; 2022b; Liang et al., 2022a; Shi et al., 2022) have found that training recurrent-based VSR models with longer video clips often leads to more accurate VSR results, as longer sequences facilitate the learning of long-range temporal dependencies. However, as the depth and temporal modeling complexity of the VSR network increases, simply extending the length of training clips becomes highly time-consuming and memory-intensive. Consequently, state-of-the-art recurrent-based VSR models are typically trained on shorter video clips to balance performance and training efficiency. This compromise, however, results in models that fail to fully learn accurate long-range temporal dependencies in long video datasets, ultimately degrading performance. Efficiently capturing long-term information propagation patterns in long video sequences during VSR model training remains a critical open challenge in the VSR field.

In this paper, we provide a feasible solution to this issue with a novel VSR training framework utilizing **L**ong-Range **R**efocused **T**emporal **I**nformation (LRTI-VSR). As the pivotal innovation, we introduce a generic training strategy for recurrent-based VSR models that efficiently learns accurate long-term temporal dependencies from long video sequences while maintaining manageable training overhead. To be more specific, since training solely on short video clips fails to learn the temporal propagation patterns inherent in long video sequences, we conduct forward and backward propagation in network training with different lengths of video clips. The forward propagation process is conducted on long video sequences to obtain accurate intermediate hidden states for all the frames, while the backward propagation step is performed on short video clips to facilitate efficient training. Extensive experiments on our proposed model and a wide range of existing recurrent-based VSR models demonstrate the effectiveness of this strategy, consistently improving performance without modifying the underlying network architectures. In addition, we rethink the role of aligned redundant temporal propagation hidden states in recovering the current frame and propose a refocused intra- and inter-frame attention structure. By replacing `SoftMax` normalization layer with a sparse refocus activation function `ReLU`$^2$ in the attention module, our model only selectively prioritizes useful information from previously processed features. To further enhance inter-frame information utilization, we also integrate the aligned hidden state from the previous frame into the FFN structure through a refocused gate unit. These improvements enable the proposed refocused intra- and inter-frame transformer block to enhance the LRTI-VSR model's ability to leverage temporal propagation features, establishing a stronger baseline for VSR tasks. Above all, our contributions can be summarized as follows.

- We propose a novel VSR training framework that efficiently learns the temporal propagation patterns of long videos and precisely utilizing inter-frame dependencies.

- We introduce a generic training strategy for recurrent-based VSR models, which enables the effective learning of accurate long-term propagation patterns while training on shorter video clips.

- We design a refocused intra&inter-frame transformer block, which selectively prioritizes useful information from redundant temporal propagation hidden states to enhance current frame recovery.

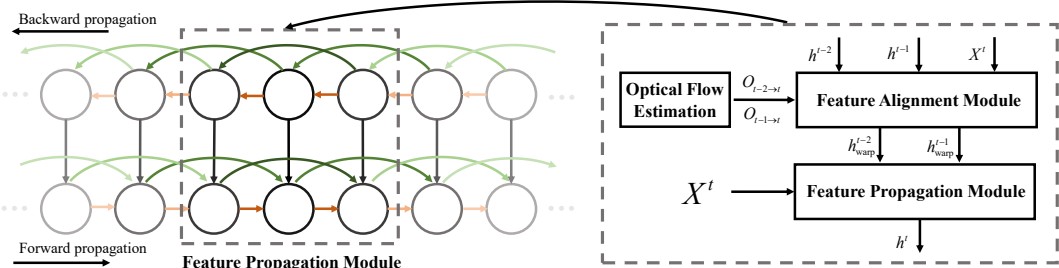

Figure 3: **Structure of feature propagation module.** The commonly used feature propagation module contains a second-order grid propagation structure (*red* and *green* solid lines) that leverages the computed hidden states $\boldsymbol{h}_{warp}^{t-2}$ and $\boldsymbol{h}_{warp}^{t-1}$ of the previous two frames warped (feature alignment module) with the current frame's feature $\boldsymbol{X}^t$ to assist in the restoration of the current frame.

- We evidence the superiority of our LRTI-VSR through extensive comparisons with state-of-the-art VSR models, achieving superior performance while maintaining manageable training overhead.

## 2 METHODOLOGY

### 2.1 PRELIMINARY

Given a low-resolution video input $\boldsymbol{I}^{LR} \in \mathbb{R}^{T \times H \times W \times 3}$ with $T$ frames, the goal of the VSR model is to reconstruct the corresponding high-resolution video $\boldsymbol{I}^{HR} \in \mathbb{R}^{T \times sH \times sW \times 3}$, where $s$ is the scaling factor and $H$, $W$, 3 are the height, width and number of channels of the input frames, respectively.

Current VSR methods can be categorized into sliding-window based and recurrent-based approaches. Unlike sliding-window based VSR methods (Li et al., 2020; Wang et al., 2019; Cao et al., 2021; Liang et al., 2022a) which rely on a limited number of adjacent frames, recurrent-based VSR methods exploit long-range temporal information, enabling more efficient and robust restoration during inference. The structure of temporal information propagation in recurrent-based VSR models (i.e. the feature propagation module) is illustrated in Fig. 3. As highlighted in previous works on recurrent-based VSR (Fuoli et al., 2019; Chan et al., 2021), the feature propagation module leverages high-dimensional hidden states computed from previous frames to assist in the computation of the current frame feature. By incorporating longer video sequences during training, the recurrent-based VSR model can more accurately learn long-range temporal propagation patterns, which can significantly improve its performance.

Furthermore, effective modeling of inter-frame information is also crucial for better VSR performance. To this end, recent studies (Chan et al., 2022a; Shi et al., 2022) have introduced second-order connection structures to extract supplementary temporal information from previous frames, enhancing inter-frame redundancy (as also shown in Fig. 3). The feature alignment module then align these hidden states from previous frames with current frame's feature by the optical flow $\boldsymbol{O}_{t-2 \to t}$ and $\boldsymbol{O}_{t-1 \to t}$ obtained from the low-resolution input for better performance. Combined with advanced inter-frame alignment techniques (Liang et al., 2022b; Shi et al., 2022; Xu et al., 2023) and cross-attention modules (Zhou et al., 2024), these approaches enable more accurate exploitation of temporal information, further advancing the state-of-the-art in video super-resolution.

### 2.2 LEARNING LONG-TERM DEPENDENCIES VIA SHORT VIDEO CLIP TRAINING

As discussed in previous subsection, temporal propagation hidden states play an important role in the performance of recurrent-based video super-resolution models. Recent studies (Chan et al., 2021; 2022b; Liang et al., 2022a; Shi et al., 2022) have shown that incorporating longer video clips into training leads to better SR results. However, since recurrent-based VSR methods typically rely on the Back-Propagation Through Time (BPTT) (Werbos, 1990) strategy, simply increasing the length of training video clips makes the training process time-consuming and memory-intensive. To address this issue, we propose a truncated backpropagation training strategy inspired by TBPTT (Williams & Zipser, 2013). Our approach enables the model to learn temporal propagation patterns from long video sequences while efficiently training on shorter clips.

**Forward propagation on long video sequences.** Before training with short video clips as input, the model performs a forward propagation on the corresponding long video sequence $\boldsymbol{I}^{LR} = \{x_1, x_2, \ldots, x_T\}$ of $T$ frames to obtain the temporal propagation hidden states $H$ for all frames:

$$\boldsymbol{H} = \text{VSR-model.forward}(\boldsymbol{I}^{LR}). \tag{1}$$

This process follows the original bidirectional recurrent-based VSR inference method, where the initial frame has no additional temporal propagation features as input.

**Backpropagation on short video clips.** A short video clip of length $L$ is selected from the long video sequence before training, which can be expressed as:

$$\boldsymbol{I}^{LR}_{clip} = \{\boldsymbol{x_t}\}^{t_{start}+L-1}_{t=t_{start}}, t_{start} \sim \text{Uniform}(1, T-L+1), \tag{2}$$

where $t_{start}$ denotes the starting frame position of the selected short video clip within the long video sequence. In practice, we additionally input the two previous and subsequent frames of the selected video clip for the bidirectional optical flow estimation:

$$\boldsymbol{O}^{LR}_{clip} = f_{optical\_flow}(\{\boldsymbol{x_t}\}^{t_{start}+L+1}_{t=t_{start}-2}), t_{start} \sim \text{Uniform}(3, T-L-1), \tag{3}$$

The bidirectional recurrent VSR model then takes the selected video clip along with the corresponding temporal propagation hidden states $\boldsymbol{H}_{clip} = \{\boldsymbol{h}^{t_{start}-2}, \boldsymbol{h}^{t_{start}-1}, \boldsymbol{h}^{t_{start}+L}, \boldsymbol{h}^{t_{start}+L+1}\}$ before and after starting and ending frames of the clip and optical flow estimation $\boldsymbol{O}^{LR}_{clip}$ as input:

$$\hat{\boldsymbol{I}}^{HR}_{clip} = \text{VSR-model}(\boldsymbol{I}^{LR}_{clip}, \boldsymbol{H}_{clip}, \boldsymbol{O}^{LR}_{clip}), \tag{4}$$

where $\hat{\boldsymbol{I}}^{HR}_{clip}$ is the output of the forward propagation process during the VSR model's training. This configuration aligns with the foundational setup of **bidirectional second-order grid propagation**, a widely adopted paradigm in state-of-the-art VSR frameworks (Chan et al., 2022a; Liang et al., 2022b; Shi et al., 2022; Xu et al., 2023; Zhou et al., 2024). With leveraging the advanced feature alignment model (Shi et al., 2022), our VSR model utilizes the aligned hidden states of long video sequences to further assist the restoration of the whole video sequence.

---

**Algorithm 1** Workflow of the proposed training strategy.

---

**Require:** training VSR-model $\boldsymbol{f_\theta}(\cdot)$
**Require:** Paired training dataset $(\boldsymbol{I}^{LR}, \boldsymbol{I}^{HR})$, Sample times $N$, Truncated video clip length $L$, Whole video length $T$
1: **while** not converged **do**
2:    Sample $\boldsymbol{I}^{LR}$                                ▷ Select a long video sequence
3:    $\boldsymbol{H} = \text{VSR-model.forward}(\boldsymbol{I}^{LR})$         ▷ Hidden states of the video sequence
4:    **for** $n = 0, 1, \cdots, N-1$ **do**
5:       Sample $t_{start} \sim \text{Uniform}(3, T-L-1)$         ▷ Select a small video clip
6:       $\boldsymbol{I}^{LR}_{clip} = \{\boldsymbol{x_t}\}^{t_{start}+L-1}_{t=t_{start}}$
7:       $\boldsymbol{O}^{LR}_{clip} = f_{optical\_flow}(\{\boldsymbol{x_t}\}^{t_{start}+L+1}_{t=t_{start}-2})$       ▷ Optical flow of the video clip
8:       $\boldsymbol{H}_{clip} = \{\boldsymbol{h}^{t_{start}-2}, \boldsymbol{h}^{t_{start}-1}, \boldsymbol{h}^{t_{start}+L}, \boldsymbol{h}^{t_{start}+L+1}\}$  ▷ Hidden states of the video clip
9:       $\hat{\boldsymbol{I}}^{HR}_{clip} = \text{VSR-model}(\boldsymbol{I}^{LR}_{clip}, \boldsymbol{H}_{clip}, \boldsymbol{O}^{LR}_{clip})$       ▷ VSR with aligned hidden states
10:      $\mathcal{L}_{sr} = \text{Charbonnier\_Loss}(\hat{\boldsymbol{I}}^{HR}_{clip}, \boldsymbol{I}^{HR}_{clip})$
11:      $\text{VSR-model.backward()}$
12:    **end for**                          ▷ This video sequence has been fully utilized
13: **end while**
14: **return** Converged VSR-model $f_\theta(\cdot)$.

---

**Implementation details.** In practice, we sample a fixed number of short clips from each long video for back-propagation. The entire workflow of the proposed training strategy is shown in Alg. 1. We intuitively ensure that the product of sampling times and truncation video clips length to be close to the total video sequence length to better balance training efficiency and VSR performance. Finally, we validate the effectiveness of our training strategy on previous CNN and Transformer-based state-of-the-art VSR models, respectively, and further evaluate the time and memory efficiency of the proposed training strategy. The detailed experimental setup and results can be seen in Section 3.

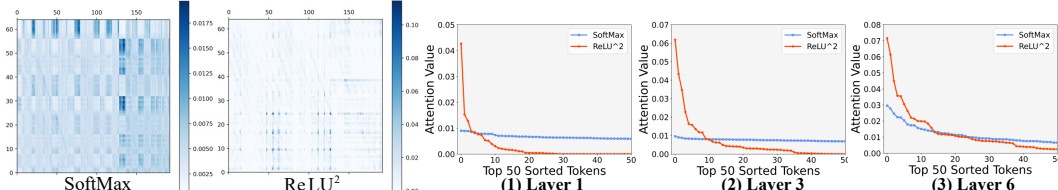

Figure 4: **Attention map comparison.** We compare the visualization of the intra&inter attention map under $\texttt{ReLU}^2$ and the original $\texttt{SoftMax}$. The Top 50 values of all tokens ($64\times192$) of the attention maps under different network depths using these two activation functions are also counted.

## 2.3 REFOCUSED INTER- AND INTRA-FRAME TRANSFORMER

When utilizing temporal propagation hidden states enhances inter-frame correlation in VSR models, inaccurate optical flow alignment will may degrade VSR model performance (Shi et al., 2022). Some methods (Shi et al., 2022; Liang et al., 2022b; Xu et al., 2023) suggest using more advanced alignment modules to alleviate this problem. To address this, we propose a **R**efocused **I**nter&intra-**F**rame **T**ransformer **B**lock (**RITB**), which extends the approach in (Zhou et al., 2024) by directly applying intra- and inter-frame attention to both previously hidden states and current frame features. Our RITB selectively prioritizes features with positive contributions from inter-frame information while mitigating the influence of irrelevant features. Specifically, we first generate Query Tokens from the current frame feature and Key/Value Tokens from both the current frame feature and hidden states of previous frames, which is same as (Zhou et al., 2024):

$$
\begin{cases}
\boldsymbol{Q}_{m,n}^t = \boldsymbol{X}_{m,n}^t \boldsymbol{W}_{m,n}^Q, \\
\boldsymbol{K}_{m,n}^t = \left[ \boldsymbol{h}_{m+1}^{t-1}; \boldsymbol{h}_{m+1}^{t-2}; \boldsymbol{X}_{m,n}^t \right] \boldsymbol{W}_{m,n}^K, \\
\boldsymbol{V}_{m,n}^t = \left[ \boldsymbol{h}_{m+1}^{t-1}; \boldsymbol{h}_{m+1}^{t-2}; \boldsymbol{X}_{m,n}^t \right] \boldsymbol{W}_{m,n}^V,
\end{cases}
\tag{5}
$$

Where $\boldsymbol{Q}_{m,n}^t$, $\boldsymbol{K}_{m,n}^t$ and $\boldsymbol{V}_{m,n}^t$ are the Query, Key and Value Tokens generated from the current input feature $\boldsymbol{X}_{m,n}^t$ and the temporal propagation hidden states $\{\boldsymbol{h}_{m+1}^{t-1}, \boldsymbol{h}_{m+1}^{t-2}\}$ of the previous frames which are aligned by the feature alignment model; $\boldsymbol{W}_{m,n}^Q$, $\boldsymbol{W}_{m,n}^K$ and $\boldsymbol{W}_{m,n}^V$ are the respective projection matrices; $\boldsymbol{m}$ and $\boldsymbol{n}$ denote the $\boldsymbol{m}$-th feature propagation module and the $\boldsymbol{n}$-th RITB block within the feature propagation module, respectively.

**Refocused attention with sparse refocus activation.** As mentioned above, to mitigate the influence of irrelevant features in aligned hidden states, we introduce the sparse refocus activation function $\texttt{ReLU}^2$ during the attention calculation:

$$
\texttt{RITB}_{\text{Attention}} = \texttt{ReLU}^2(\boldsymbol{Q}_{m,n}^t \boldsymbol{K}_{m,n}^t{}^T / \sqrt{d} + B) \boldsymbol{V}_{m,n}^t,
\tag{6}
$$

Table 1: The effects of different activation functions after the attention map.

| Activation Function | PSNR(dB) |
|---|---|
| $\texttt{SoftMax}$ | 30.83 |
| $\texttt{SoftMax}$ w/o negative similarity | 30.92 |
| **$\texttt{ReLU}^2$** | **31.02** |

where $d$ is the channel dimension of the token, and $B$ is the learnable relative positional encoding. Unlike traditional transformer blocks that use $\texttt{SoftMax}$ to retain all feature correlations, we employ the $\texttt{ReLU}^2$ function to set negative values in the $\boldsymbol{Q} \times \boldsymbol{K}$ matrix to zero while amplifying positive correlation values. This leads to more effective utilization of inter-frame temporal information, as illustrated in Fig. 4 and Table 1. By setting negative attention values to zero and enhancing positive ones with $\texttt{ReLU}^2$, we achieved consistent performance gains.

**Refocused gated unit (RGU).** Beyond the attention module, we further leverage inter-frame temporal information within the Feed-Forward Networks (FFN) module by introducing a **R**efocused **G**ated **U**nit (**RGU**). Specifically, we leverage the aligned temporal propagation hidden state $\boldsymbol{h}_{m+1}^{t-1}$ of the previous frame for FFN calculation after calculating the refocused intra&inter-frame attention to enhance the feature $\boldsymbol{X}_{m,n}^t{}'$ of the current frame:

$$
\texttt{RITB}_{\text{FFN}} = g(\texttt{ReLU}^2(f(\boldsymbol{X}_{m,n}^t{}')) \odot f(\boldsymbol{h}_{m+1}^{t-1})),
\tag{7}
$$

Where $f(\cdot)$ and $g(\cdot)$ are linear projections, $\odot$ is indicates element-wise multiplication and we also used $\texttt{ReLU}^2$ as the non-linear activation function. Additionally, our transformer layer incorporates LayerNorm, a common component in Transformer-based architectures (Liang et al., 2022b; Shi et al., 2022; Xu et al., 2023; Zhou et al., 2024). The overall structure of the proposed refocused intra&inter-frame transformer block (RITB) is illustrated in Fig. 5.

## 2.4 MODEL IMPLEMENTATIONS

In this subsection, we detail the implementation of our proposed LRTI-VSR model. Our LRTI-VSR framework is built upon the bi-directional second-order grid propagation framework of BasicVSR++ (Chan et al., 2022a), which has also been adopted in recent state-of-the-art methods (Shi et al., 2022; Xu et al., 2023; Zhou et al., 2024). The whole model consists of three parts, i.e. the shallow feature extraction part, the recurrent feature refinement part and the feature reconstruction part. Generally, the recurrent feature refinement part comprises **M** feature propagation modules and each feature propagation module consists of **N** cascaded processing RITB blocks. We follow previous works (Chan et al., 2022a; Shi et al., 2022; Xu et al., 2023; Zhou et al., 2024) which use a plain convolution operation to extract shallow features and adopt a pixel-shuffle layer (Shi et al., 2016) to reconstruct HR output with refined features. The patch alignment method used in PSRT (Shi et al., 2022) is used to align the hidden states with the current frame. Following previous state-of-the-art approaches, we utilize the Charbonnier loss (Char-

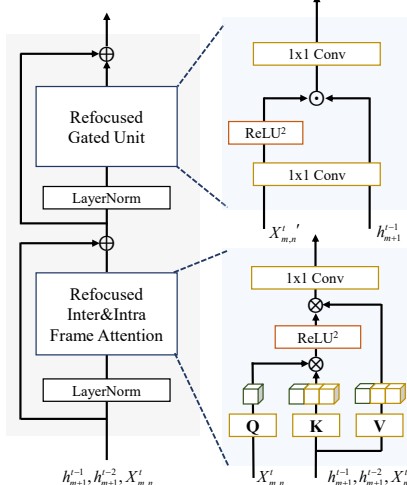

Figure 5: **Illustration of the refocused intra&inter frame transformer block (RITB).** More details of our RITB block can be found in Subsection 2.3.

bonnier et al., 1994) $\mathcal{L}_{sr} = \sqrt{\| \hat{\boldsymbol{I}}^{HQ} - \boldsymbol{I}^{HQ} \|^2 + \varepsilon^2} (\epsilon = 10^{-3})$ between the estimated HR image $\hat{\boldsymbol{I}}^{HQ}$ and the ground truth image $\boldsymbol{I}^{HQ}$ to train our network in all of our experiments. The detailed overall architecture of the proposed LRTI-VSR model and structural comparison of the proposed RITB block with MFSAB and IIAB adapoted in PSRT (Shi et al., 2022) and MIA-VSR (Zhou et al., 2024) are shown in the **Appendix A.2**. Furthermore, the specific settings of our LRTI-VSR model in the various comparison experiments will be introduced in Section 3.

## 3 EXPERIMENTS

### 3.1 EXPERIMENTAL SETTINGS

To evaluate the ability of our proposed LRTI-VSR model to learn long-term dependencies with short video clips, we conduct extensive experiments on the REDS (Nah et al., 2019) and ToS3 (Chu et al., 2020) datasets. The REDS dataset is a widely-used video dataset with each video sequence contains 100 frames, and the ToS3 dataset is the test set of 3 long video sequences (room,bridge,and face) of lengths 150, 166 and 233. We train our LRTI-VSR model on the REDS dateset with bicubic downsampling from scratch for 600K iterations, and evaluate all the proposed VSR models on the REDS testing data (i.e. REDS4) and ToS3 datatset. In addition, we also validate our LRTI-VSR model on the real-world VideoLQ (Chan et al., 2022b) dataset, and the experimental visual results are in the **Appendix A.4**. We implement our model with PyTorch and train and test all of our models with RTX 4090 GPUs. The respective hyper-parameters used for ablation studies and comparison with state-of-the-art methods will be introduced in the following subsections and **Appendix A.5**.

### 3.2 ABLATION STUDY

**The effectiveness of proposed training strategy.** In order to demonstrate the effectiveness of our proposed truncated backpropagation (denoted as TB) training strategy, we first apply it to several previous state-of-the-art CNN-based and Transformer-based VSR models, e.g. BasicVSR(Chan et al., 2021), BasicVSR++(Chan et al., 2022a) and PSRT(Shi et al., 2022), respectively. For CNN-based VSR methods BasicVSR and BasicVSR++, we follow their **original basic training setups** and train them using our proposed TB training strategy (truncated length 15 frames, sampling 7 times and truncated length 30 frames, sampling 5 times). For the Transformer-based PSRT method, we perform 100K iterations on the pre-trained model using the proposed TB training strategy

(truncation length 16 frames, sampling 6 times) to enhance efficiency. The VSR results of these models on the REDS4 (Nah et al., 2019) dataset are shown in Table 2. We also report the baseline training time (gray font) and the training time required after applying our proposed TB training strategy in this table. With only marginally increased training costs, our proposed training strategy improves the PSNR of the CNN-based VSR models

Table 2: Ablation studies of our proposed truncated backpropagation strategy (denoted as *TB*) on previous state-of-the-art CNN&Transformer VSR models.

| Methods | Training Length | REDS4 PSNR | SSIM | GPU days |
|---|---|---|---|---|
| BasicVSR(Chan et al., 2021) | 15 | 31.42dB | 0.8909 | 31.2 |
| +*TB strategy* | 15 | **31.56dB** | **0.9060** | 35.2 |
| BasicVSR++(Chan et al., 2022a) | 30 | 32.39dB | 0.9069 | 57.6 |
| +*TB strategy* | 30 | **32.49dB** | **0.9176** | 60.8 |
| PSRT(Shi et al., 2022) | 16 | 32.72dB | 0.9106 | 235 |
| +*TB strategy* | 16 | **32.84dB** | **0.9234** | +21.2 |

BasicVSR and BasicVSR++ by 0.14dB and 0.1dB, respectively; and for the previous state-of-the-art PSRT model, our TB training strategy improves the performance of its VSR results by 0.12dB. These results highlight the generalizability of our training strategy across different VSR architectures.

Table 3: Ablation studies on the effectiveness of our proposed truncated backpropagation training strategy (TB).

| Method | TB | Training Length | Memory | GPU days | REDS4 PSNR | SSIM |
|---|---|---|---|---|---|---|
| | | 8 | 3.9GB | 3.33 | 31.08dB | 0.8987 |
| **Ours** | √ | 8 | **3.8GB** | **3.75** | **31.44dB** | **0.9052** |
| | | 24 | 10.9GB | 9.12 | 31.44dB | 0.9045 |
| | | 40 | 18.4GB | 15.40 | 31.61dB | 0.9079 |

Table 4: Ablation studies on different truncated lengths for training.

| Truncated Length | Total Length | Memory | GPU days | PSNR |
|---|---|---|---|---|
| - | 8 | 3.9GB | 3.33 | 31.08dB |
| 4 | 40 | 2.9GB | 2.83 | 31.22dB |
| **8** | **40** | **3.8GB** | **3.75** | **31.44dB** |
| 16 | 40 | 4.1GB | 8.62 | 31.50dB |
| - | 40 | 18.4GB | 18.4 | 31.61dB |

Furthermore, to evaluate the training efficiency of our proposed training strategy for long video sequences, we separately instantiate our LRTI-VSR model and train 300K iterations from scratch. We use 6 RITB blocks to to build feature propagation modules and instantiate our LRTI-VSR model with 4 feature propagation modules. Concretely, we set different training lengths of video clips to compare the efficiency of using only truncated length 8 frames with our proposed training strategy (sampling 5 times within 40 frame-length video clips). We also report the GPU memory consumption and the number of GPU days required for each experiment. To ensure a fair comparison, all these models are trained with **batch size of 1 and evaluated on one RTX4090 GPU**. As shown in Table 3, the proposed truncated back-propagation training strategy achieves a performance comparable to training with video clips of length 24 when using video clips truncated to a length of 8, while attaining an acceleration of nearly **2.5×** in training time and a reduction of nearly **2.9×** in GPU memory consumption. In Table 4, we also report the results for different truncated lengths (sampling 5 times within 40 frame-length video clips). Since our training strategy loads new video sequences only after a certain number of iterations,, our training strategy can effectively reduce the amount of VRAM required during training. Additionally, our training strategy incurs only an acceptable increase in training time compared to the baseline approach of training with video clips of length 8, but yields a substantial performance improvement of **0.36dB**.

**The effectiveness of RITB.** In this part, we validate the effectiveness of the proposed refocused intra&inter transformer block (RITB). We firstly compare the RITB with the multi-frame self-attention block (MFSAB) and intra&inter-frame attention block (IIAB) which were adopted in PSRT (Shi et al., 2022) and MIA-VSR (Zhou et al., 2024). As mentioned above, we also use 6 MFSAB or IIAB or RITB to build feature propagation modules and instantiate VSR models with 4 feature propagation modules and also evaluate these models on **one RTX4090 GPU**.

Table 5: Ablation studies on the proposed RITB block. More details can be found in the subsection 3.2.

| Methods | ReLU² | RGU | TB | Params (M) | REDS4 PSNR | SSIM |
|---|---|---|---|---|---|---|
| MFSAB(Shi et al., 2022) | | | | 6.41 | 30.76 | 0.8918 |
| IIAB(Zhou et al., 2024) | | | | 6.06 | 30.83 | 0.8929 |
| | √ | | | 6.06 | 31.02 | 0.8969 |
| **RITB** | √ | √ | | 6.06 | 30.97 | 0.8976 |
| | √ | √ | | 6.06 | 31.08 | 0.8987 |
| | √ | √ | √ | 6.06 | **31.44** | **0.9052** |

In addition, we perform ablation experiments on the improvements proposed in RITB respectively to verify the effectiveness of each part of RITB. As can be found in Table 5, our model achieves a 0.19dB improvement over the baseline IIAB-VSR model with the addition of only the ReLU² activation function, and a 0.14dB improvement when only the Refocused Gated Unit (RGU) is added. When combining both modifications, our LRTI-VSR model with the RITB block outper-

Table 6: Quantitative comparison (PSNR/SSIM) on the REDS4 (Nah et al., 2019) and ToS3 (Chu et al., 2020) datasets for 4× video super-resolution task. The number of FLOPs(T) are computed on an LR frame size of $180 \times 320$. For all experiments, we color the best performance with red.

| Method | Training Length | Params(M) | FLOPs(T) | REDS4 PSNR | REDS4 SSIM | ToS3 PSNR | ToS3 SSIM |
|---|---|---|---|---|---|---|---|
| TOFlow(Xue et al., 2019) | 5 | - | - | 27.98 | 0.7990 | - | - |
| EDVR(Wang et al., 2019) | 5 | 20.6 | 2.95 | 31.09 | 0.8800 | 32.96 | 0.9100 |
| MuCAN(Li et al., 2020) | 5 | - | - | 30.88 | 0.8750 | - | - |
| VSR-T(Cao et al., 2021) | 5 | 32.6 | 1.60 | 31.19 | 0.8815 | 33.11 | 0.9125 |
| VRT(Liang et al., 2022a) | 6 | 35.6 | 1.37 | 31.60 | 0.8888 | 33.68 | 0.9217 |
| BasicVSR(Chan et al., 2020) | 15 | 6.3 | 0.33 | 31.42 | 0.8909 | 32.51 | 0.9015 |
| IconVSR(Chan et al., 2020) | 15 | 8.7 | 0.51 | 31.67 | 0.8948 | 33.69 | 0.9260 |
| BasicVSR++(Chan et al., 2022a) | 30 | 7.3 | 0.39 | 32.39 | 0.9069 | 34.24 | 0.9339 |
| LRTI-VSR-small | 16 | 6.06 | 0.58 | 32.56 | 0.9084 | 34.37 | 0.9342 |
| VRT(Liang et al., 2022a) | 16 | 35.6 | 1.37 | 32.19 | 0.9006 | 34.09 | 0.9289 |
| RVRT(Liang et al., 2022b) | 30 | 10.8 | 2.21 | 32.75 | 0.9113 | 34.47 | 0.9354 |
| PSRT-recurrent(Shi et al., 2022) | 16 | 13.4 | 2.39 | 32.72 | 0.9106 | 34.48 | 0.9361 |
| MIA-VSR(Zhou et al., 2024) | 16 | 16.5 | 1.61 | 32.78 | 0.9115 | 34.12 | 0.9330 |
| IART(Xu et al., 2023) | 16 | 13.4 | 2.51 | 32.90 | 0.9138 | 34.63 | 0.9386 |
| LRTI-VSR | 16 | 12.9 | 1.54 | 33.06 | 0.9162 | 34.81 | 0.9399 |

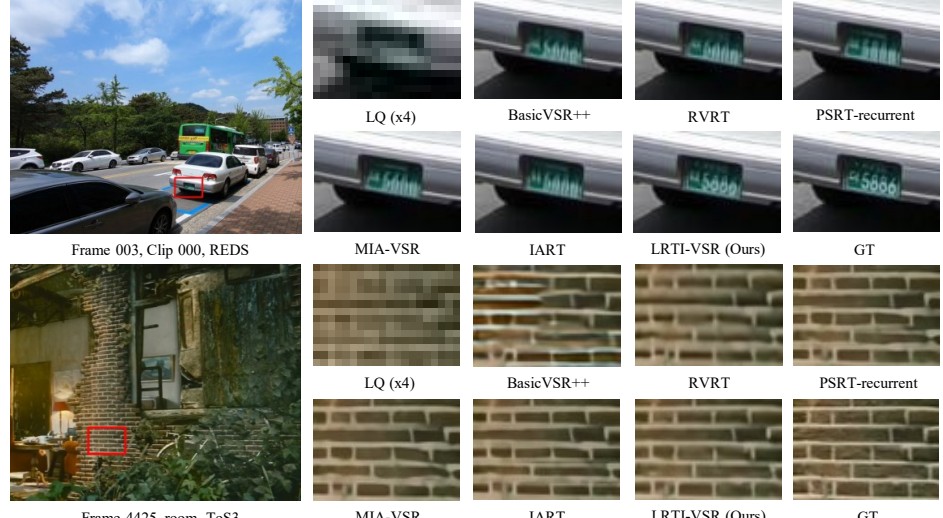

Figure 6: Visual comparison on REDS4 (Nah et al., 2019) and ToS3 (Chu et al., 2020) dataset.

forms the IIAB-VSR model by **0.24dB** with no additional parameter increase. Furthermore, with the integration of our proposed truncated backpropagation training strategy, the LRTI-VSR model demonstrates a performance improvement by a large margin of **0.61dB** and **0.68dB** over the baseline IIAB-VSR (Zhou et al., 2024) and MFSAB-VSR (PSRT) (Shi et al., 2022) model. Furthermore, for the model used in our ablation study (named LRTI-VSR-small in Table 6), we trained it from scratch using the basic experimental setup of BasicVSR++ (Chan et al., 2022a) with a shorter training length (16 **vs.** 30), resulting in a performance improvement of over **0.17 dB** compared to BasicVSR++.

## 3.3 COMPARISON WITH STATE-OF-THE-ART METHODS

In this subsection, we compare the proposed LRTI-VSR model with current state-of-the-art VSR methods. We compare the proposed LRTI-VSR with representative sliding-window based methods TOFlow (Xue et al., 2019), EDVR (Wang et al., 2019), MuCAN (Li et al., 2020), VSR-T (Cao et al., 2021), VRT (Liang et al., 2022a), RVRT (Liang et al., 2022b) and recurrent-based methods BasicVSR (Chan et al., 2020), BasicVSR++ (Chan et al., 2022a), PSRT (Shi et al., 2022), IART(Xu

et al., 2023) and MIA-VSR(Zhou et al., 2024); among which VRT, RVRT, PSRT-recurrent, IART and MIA-VSR are Transformer-based approaches and the other approaches are CNN-based models.

In order to compare with state-of-the-art methods, we instantiate our LRTI-VSR model with 4 feature propagation modules and each feature propagation module contains 18 RITB blocks. Among them, we set the interval of skip connections to [6,6,6]. The spatial window size, head size and channel size are set to $8 \times 8$, 6 and 120 accordingly. The number of parameters in our model is on par with the recent state-of-the-art methods PSRT-recurrent (Shi et al., 2022) and IART(Xu et al., 2023). In particular, we use our proposed truncated backpropagation training strategy for our LRTI-VSR model, which the truncated video clip length is set to 16 and then sampling 6 times within the whole video (100 frames, REDS).

The VSR results of different methods on the long-video test datasets can be found in Table 6. Our model improves the PSNR of the baseline MIA-VSR by 0.28dB on the REDS dataset and 0.69dB on the ToS3 dataset, while having a lower number of parameters and FLOPs. Furthermore, our model improves the PSNR of the state-of-the-art IART model by **0.16dB** on the REDS dataset and **0.18dB** on the ToS3 dataset, while the number of FLOPs (test on the REDS4 dataset, $180 \times 320$ in size) is nearly **40%** lower. Some visual results from different VSR results can be found in Fig.6, our LRTI-VSR method is able to recover more natural and sharp textures from the input LR sequences. More visual examples are also provided in our in the **Appendix A.7**.

## 4 RELATED WORK

We discuss with the most relevant literature here and provide a more discussion in Appendix A.6.

**Recurrent-based Video Super-Resolution Models.** Compared with Temporal sliding-window methods (Li et al., 2020; Wang et al., 2019; Cao et al., 2021; Liang et al., 2022a) that only use short-range temporal information, another category of approaches applies recurrent neural networks to exploit long-range temporal information from more frames during inference. BasicVSR (Chan et al., 2020) utilized bi-directional hidden states, and BasicVSR++ (Chan et al., 2022a) further improved BasicVSR with second-order grid propagation and flow-guided deformable alignment. Recently, more advanced inter-frame alignment modules (e.g. RVRT (Liang et al., 2022b), PSRT (Shi et al., 2022) and IART (Xu et al., 2023)) and computationally efficient network layers (e.g. MIA-VSR (Zhou et al., 2024)) have been proposed to improve the utilization efficiency of inter-frame information. Combined with the advanced Transformer architecture, the performance of VSR is improved to a new level. Our proposed LRTI-VSR model follows the general framework of existing transformer-based recurrent VSR models but leverages long-term dependencies within long video sequences with affordable training overhead.

**Efficient Training Strategy of RNNs.** Training RNNs often relies on the resource-intensive Back-Propagation Through Time (Werbos, 1990) (BPTT) method. To address its computational challenge, Truncated Back-Propagation Through Time (Williams & Zipser, 2013) (TBPTT) was originally designed for recurrent neural networks to model long natural language sequences. PGT (Pang et al., 2021) is the first attempt to introduce the TBPTT strategy into video modeling in high-level vision tasks. In the VSR field, Lin et al. (2023) proposed to accelerate training by gradually increasing the video resolution and frame length, but it was still limited by peak memory usage. Our proposed training strategy bears conceptual similarity to TBPTT. It not only maintains the original training cost but also further facilitates its learning of long-term dependencies of long video sequences.

## 5 CONCLUSION

In this paper, we proposed a novel recurrent video super-resolution training framework which leverages long-range refocused temporal information, named LRTI-VSR. We develop a generic training strategy for the recurrent-based VSR model that effectively learns the accurate temporal propagation patterns within long video sequences and facilitates training using shorter video clips. Further more, a refocused intra&inter-frame Transformer block is proposed to select and refocus features with positive contributions from the previously hidden states for current frame restoration. We evaluated our LRTI-VSR model on various benchmark video super-resolution datasets, and our model is able to achieve state-of-the-art video super-resolution results.

## REPRODUCIBILITY STATEMENT

We provide detailed hyperparameter settings in **Section 3** and **Appendix A.5**. To further facilitate reproducibility, we will release our implementation and trained model checkpoints, enabling the reported results to be reproduced.

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

# A APPENDIX

In this file, we provide more implementation and experimental details which are not included in the main text. In Section A.1, We provide an explanation in the paper regarding the use of large language models (LLMs). In Section A.2, we provide a detailed diagram of the network architecture for the entire LRTI-VSR model and structural comparison of the proposed RITB block with previous work. In Section A.3, we provide the complexity analysis of our proposed LRTI-VSR model with previous advanced Video super-resolution methods. In Section A.4, we provide additional validation experiment of the LRTI-VSR model under the real-world video dataset. In Section A.5, we provide more implementation details and more information about the dataset. In Section A.6, We provide a more detailed discussion on the related work of our proposed LRTI-VSR model. In Section A.7, we provide more visual examples of our proposed LRTI-VSR model.

## A.1 THE USE OF LARGE LANGUAGE MODELS (LLMS)

We used large language models (LLMs) to aid in polishing the writing. Specifically, LLMs were employed to improve grammar, clarity, and readability of the manuscript. No part of the research ideation, methodological design, or experimental analysis relied on LLMs.

## A.2 MODEL STRUCTURE

The overall architecture of our proposed LRTI-VSR and structural comparison of the proposed RITB block with MFSAB (Shi et al., 2022) and IIAB (Zhou et al., 2024) are shown in Figure 8 and Figure 9. We built our LRTI-VSR framework upon the bi-directional second-order grid propagation framework of BasicVSR++ (Chan et al., 2022a) and the attention structure of MIA-VSR (Zhou et al., 2024). Besides the commonly used shallow feature extraction, recurrent feature refinement and feature reconstruction parts, we utilize the proposed truncated backpropagation training method and refocused intra&inter Transformer block (RITB) to train and build our LRTI-VSR model.

## A.3 COMPLEXITY ANALYSIS

Table 7: Comparison of model size and complexity of different state-of-the-art VSR models on the REDS(Nah et al., 2019) dataset.

| Model | Params(M) | FLOPs(T) | Runtime(ms) | PSNR(dB) |
|---|---|---|---|---|
| RVRT*(Liang et al., 2022b) | 10.8 | 2.21 | 473 | 32.75 |
| PSRT(Shi et al., 2022) | 13.4 | 2.39 | 1041 | 32.72 |
| MIA-VSR(Zhou et al., 2024) | 16.5 | 1.61 | 822 | 32.78 |
| IART(Xu et al., 2023) | 13.4 | 2.51 | 1080 | 32.90 |
| LRTI-VSR | 12.9 | **1.54** | 848 | **33.06** |

\* means that uses customized CUDA kernels for better performance.

In Table 7, we report the number of parameters, the number of FLOPs, the Runtime and the PSNR on the REDS dataset by different current state-of-the art Transformer-based VSR methods. Our LRTI-VSR method has a similar or smaller number of parameters but requires the least number of FLOPs when processing video sequences. As for the runtime, our model is not as fast as RVRT, because the authors of RVRT have implemented the key components of RVRT with customized CUDA kernels. As the acceleration and optimization of transformers still require further research, there is room for further optimization of the runtime of our method by our relatively small FLOPs.

## A.4 EVALUATE ON THE REAL-WORLD DATASET

Real-world VSR is a variant of the VSR task where the low-resolution inputs are corrupted with different non-deterministic degradation. The classical VSR method usually has yield excessively smoothed results on this kind of dataset. To further prove that our proposed truncated back-propagation training method is also suitable for VSR models trained for real scenarios, we applied it

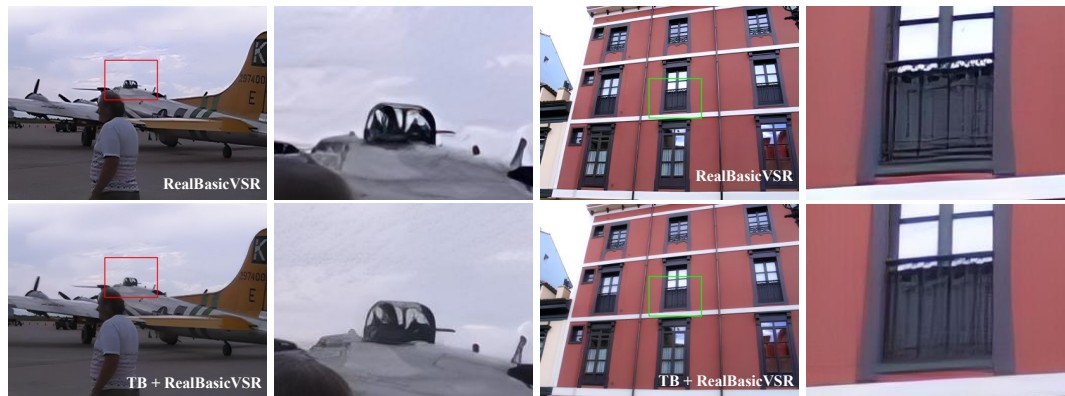

Figure 7: **Qualitative comparison on VideoLQ (Chan et al., 2022b) dataset.** Our proposed TB training strategy recovers the aircraft textures and reduces the curtain artifacts, which RealBasicVSR does not recover.

to the RealBasicVSR (Chan et al., 2022b) model. It is obvious in the Fig. 7 that our training method can help produce more realistic and fine-grained results.

### A.5 DATASET AND IMPLEMENTATION DETAILS

#### A.5.1 DATASETS

**REDS (Nah et al., 2019)** REDS is a widely-used video dataset for evaluating video restoration tasks. It has 270 clips with a spatial resolution of $1280 \times 720$. We follow the experimental settings of (Chan et al., 2020; 2022a; Shi et al., 2022) and use REDS4 (4 selected representative clips, i.e., 000, 011, 015 and 020) for testing and training our models on the remaining 266 sequences.

**ToS3 (Chu et al., 2020)** ToS3 is a long video test dataset for evaluating video super-resolution tasks. It contains 3 video clips (i.e, room, bridge and face) and the length of each video clip is 150, 166 and 233 ($534 \times 1280$). We follow the experimental settings of (Chan et al., 2020; 2022a; Shi et al., 2022; Xu et al., 2023; Zhou et al., 2024) to train the LRTI-VSR model on the REDS datatset and use the 3 sequences in the ToS3 dataset to evaluate current state-of-the-art VSR models.

**VideoLQ (Chan et al., 2022b)** VideoLQ is a test dataset with non-deterministic degradation for evaluating real-world VSR tasks. It contains 50 video clips and the length of each video clip is 100. We follow the experimental settings of (Chan et al., 2022b) to train RealBasicVSR model with our proposed TB training strategy on the REDS datatset and use the VideoLQ dataset to compare with the original RealBaiscVSR model.

#### A.5.2 TRAINING AND TESTING DETAILS

**Comparison with State-of-the-Art VSR Methods in Table 6.** We train our LRTI-VSR model with the REDS (Nah et al., 2019) training dataset with zooming factor 4. We follow the experimental settings of BasicVSR++ (Chan et al., 2022a) and train our LRTI-VSR and LRTI-VSR-small models for 600K iterations. The initial learning rate is set as $2 \times 10^{-4}$ and a cosine learning rate decay to 1e-7. We train our model with Adam optimizer and the batch size is set as 24. In the testing phase, we evaluate LRTI-VSR model's performance on the REDS4 (Nah et al., 2019) dataset and ToS3 (Chu et al., 2020) dataset. We calculate PSNR and SSIM on the RGB channel for these datatsets. For the calculation of the FLOPs in Table 4, we compare all VSR models with an input low-resolution (LR) frame size of $180 \times 320$ on the REDS4 dataset.

**Ablation Studies of Proposed Truncated Backpropagation Training Strategy in Table 2, Table 3 and Table 4 .** For the ablation study of the truncated backpropagation (TB) training strategy, we follow the original training settings of the BasicVSR (Chan et al., 2021) and BasicVSR++ (Chan et al., 2022a) models, respectively. Both models are trained from scratch using our proposed TB

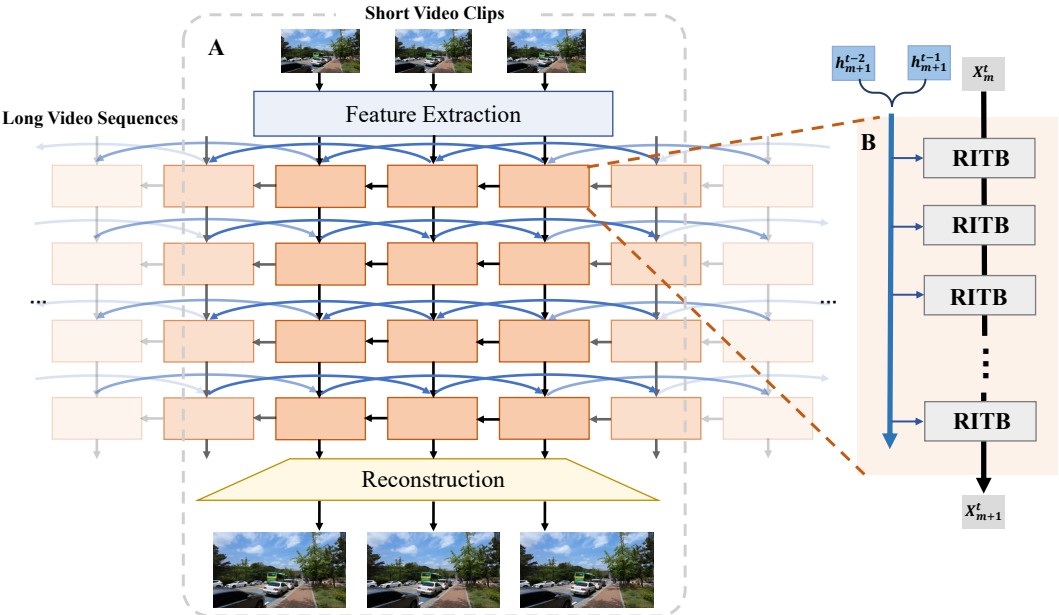

Figure 8: **The overall architecture of LRTI-VSR.** We develop a truncated backpropagation training strategy for the VSR model (**A**) learning long-term propagation patterns within long video sequences from short video clips and further proposed a refocused intra&inter Transformer block (RITB) used in the LRTI-VSR architecture to selectively prioritize useful information and suppress irrelevant features from temporal propagation hidden states to recover the current frame (**B**).

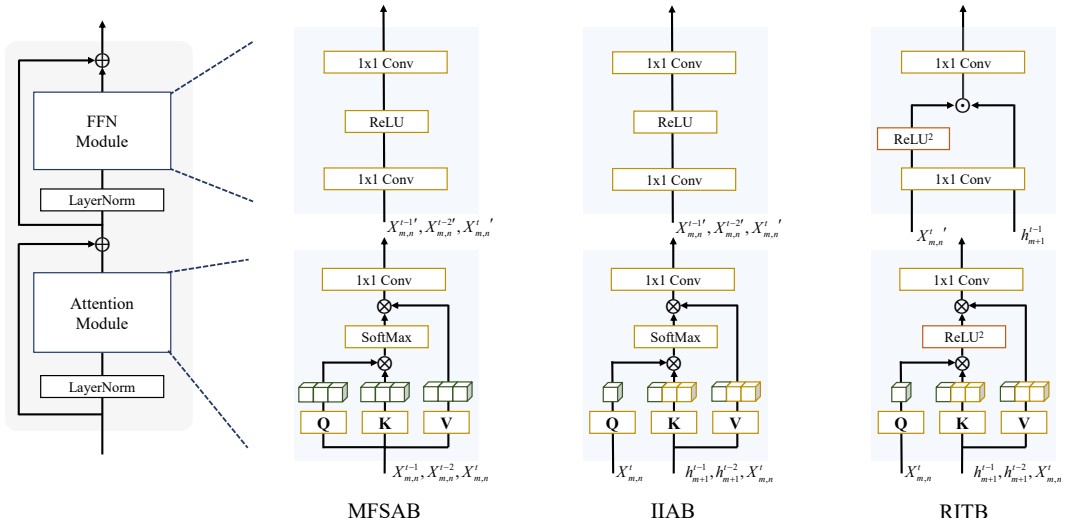

Figure 9: **The structural comparison of MFSAB, IIAB and RITB.** Our proposed a refocused intra&inter Transformer block (RITB) improves the previous MFSAB(Shi et al., 2022) block and IIAB(Zhou et al., 2024) block in two aspects: **(1)** We replace `SoftMax` with the sparse refocus activation function `ReLU`$^2$ to suppress irrelevant features with negative impacts on the current frame's recovery while refocusing on beneficial features in the cross-attention module; **(2)** We integrate the aligned hidden state from previous frame into the FFN structure through a refocused gate unit, further improving inter-frame information utilization.

training strategy. For the transformer-based method PSRT (Shi et al., 2022), we fine-tune the pretrained model with an initial learning rate of 1e-4 with TB training strategy. The learning rate was decayed to 1e-7 using a cosine schedule over 100K iterations with batchsize of 4.

For Table 3 and Table 4, we also instantiate the LRTI-VSR model with the bi-directional second-order grid propagation strategy. The total training iterations are set to 300,000, with a learning rate initialized at 1e-4 and subjected to a cosine learning rate decay, reaching 1e-7 at the end of training. For efficiency, the batch size used for these experiments is 1, respectively. All models are trained and evaluated on the REDS dataset.

**Ablation Studies of Proposed Refocused Intra&inter Transformer Block (RITB) in Table 1 and Table 5.** For ablation studies of sparse refocus activation function `ReLU`$^2$, we instantiate the the baseline IIAB model (Zhou et al., 2024) in the same configuration as the previous ablation studies in Table 3 with vanilla `SoftMax` activation function. For ablation studies of refocused intra&inter Transformer block in Table 5, we instantiate the LRTI-VSR model in the same configuration as the previous ablation studies in Table 3. The total number of training iterations of these experiments are both set to 300,000, with the learning rate initialized at 1e-4 and subject to cosine learning rate decay, reaching 1e-7 at the end of training.

## A.6    Related Work

### A.6.1    Video Super-Resolution

According to how the temporal information is utilized, all previous deep learning-based VSR methods can be divided into two categories: temporal sliding-window based methods and recurrent based methods.

Temporal sliding-window methods tend to restore a single frame using several neighboring reference frames within a temporal window, such as estimating HR images from multiple input images in a temporal sliding window manner (Li et al., 2020; Wang et al., 2019; Cao et al., 2021; Liang et al., 2022a). The alignment module plays an essential role in temporal sliding-window based method in modeling the inter-frame relationship. In the earlier stage, several sliding-window methods (Caballero et al., 2017; Liu et al., 2017; Tao et al., 2017) explicitly estimated the optical flow to align adjacent frames. Dynamic filters (Jo et al., 2018), deformable convolutions (Dai et al., 2017; Tian et al., 2020; Wang et al., 2019) and attention modules (Isobe et al., 2020b; Li et al., 2020) have been developed to conduct motion compensation implicitly in the feature space. Although the alignment module allows sliding-window-based VSR networks to better exploit temporal information from adjacent frames, the accessible temporal information is constrained by the window size, limiting the models' ability to utilize data from only a small number of input frames.

Compared with Temporal sliding-window methods that only use short-range temporal information, another category of approaches applies recurrent neural networks to exploit long-range temporal information from more frames. FRVSR (Sajjadi et al., 2018) first proposed a recurrent framework that utilizes optical flow to align the previous HR estimation and the current LR input for VSR. RLSP (Fuoli et al., 2019) propagates high-dimensional hidden states instead of the previous HR estimation to better exploit long-term information. RSDN (Isobe et al., 2020a) further extended RLSP (Fuoli et al., 2019) by decomposing the LR frames into structure and detail layers and introduced an adaptation module to selectively use the information from hidden states. BasicVSR (Chan et al., 2020) utilized bi-directional hidden states, and BasicVSR++ (Chan et al., 2022a) further improved BasicVSR with second-order grid propagation and flow-guided deformable alignment. Recently, more advanced inter-frame alignment modules (e.g. RVRT (Liang et al., 2022b), PSRT (Shi et al., 2022) and IART (Xu et al., 2023)) and computationally efficient network layers (e.g. MIA-VSR (Zhou et al., 2024)) have been proposed to improve the utilization efficiency of inter-frame information. Combined with the advanced Transformer architecture, the performance of VSR is improved to a new level. Our proposed LRTI-VSR model follows the general framework of existing transformer-based recurrent VSR models but leverages long-term dependencies within long video sequences with affordable training overhead.

### A.6.2    Efficient Modeling of Long Sequences

How to incorporate long sequence dependencies into training efficiently has always been a key issue in sequence model research. Training RNNs often relies on the resource-intensive Back-Propagation Through Time (Werbos, 1990) (BPTT) method. To address its computational challenge, Truncated Back-Propagation Through Time (Williams & Zipser, 2013) (TBPTT) was originally designed for

recurrent neural networks to model long natural language sequences. In later stages, researchers proposed unbiased approximations like NoBackTrack (Ollivier et al., 2015) and UORO (Tallec & Ollivier, 2017a), which update model parameters online and avoid the memory and computational overhead caused by backpropagation over time, facilitating efficient training of sequence models. ARTBP (Tallec & Ollivier, 2017b) utilizes a flexible memory method and compensatory factors to mitigate noise while maintaining accuracy and efficiency for long sequences. Most recently in the research of large language model, in order to make the Transformer-based sequence model able to use the information of long sequences for training, sparse Transformer (Child et al., 2019; Beltagy et al., 2020; Ding et al., 2023), compressed memory (Liu et al., 2018), KV cache (Shazeer, 2019; Dao et al., 2022; Kwon et al., 2023), linear Transformer (Katharopoulos et al., 2020) and sequence parallelism (Li et al., 2021; Gu et al., 2021) strategies are usually used for efficient long sequence modeling. PGT (Pang et al., 2021) is the first attempt to introduce the TBPTT strategy into video modeling in high-level vision tasks. However, efficiently incorporating long-term information into video restoration model training remains a critical challenge in the field of low-level vision. To the best of our knowledge, our study is the first work in this field that utilizes long-term propagation patterns of long video sequences to assist training with short video clips, requiring a minor increase in training overhead.

## A.7 VISUAL RESULTS

We conducted further visual comparisons between existing VSR methods and the proposed Transformaer-based recurrent video super-resolution framework utilizing long-range refocused temporal information (LRTI-VSR). The LRTI-VSR model is trained on the REDS dataset using a truncated sequence length of 16 frames. Figure 10 and Figure 11 illustrate the visualization results. As observed, the proposed method not only improves quantitative performance but also produces images with sharp edges, fine details, and visually appealing quality, such as the edge details of buildings, license plate numbers, and intricate details in cinematic scenes. In contrast, existing methods suffer from texture distortions or loss of detail in these scenarios.

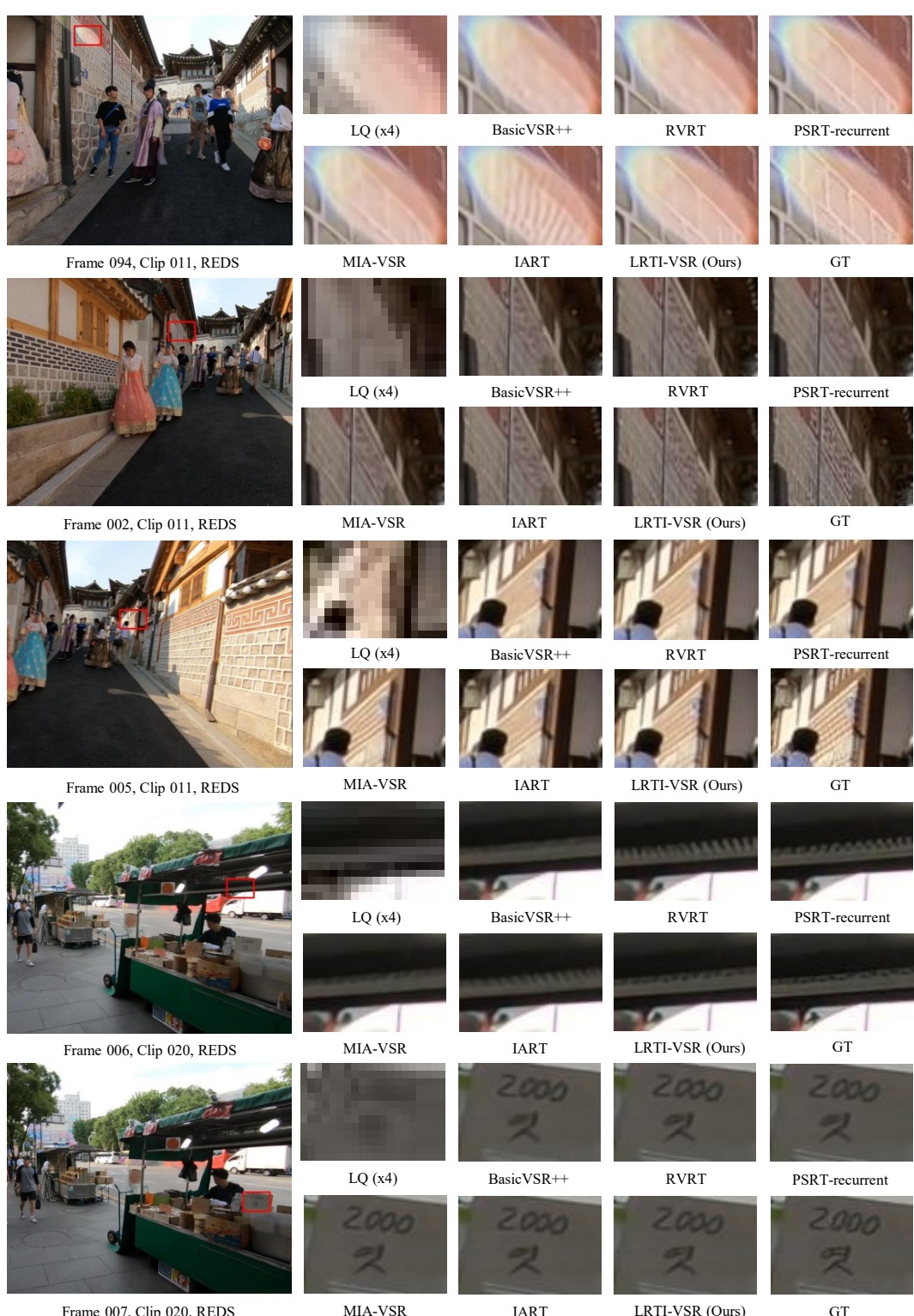

Figure 10: Visual comparison for $4\times$ VSR on REDS4 dataset.

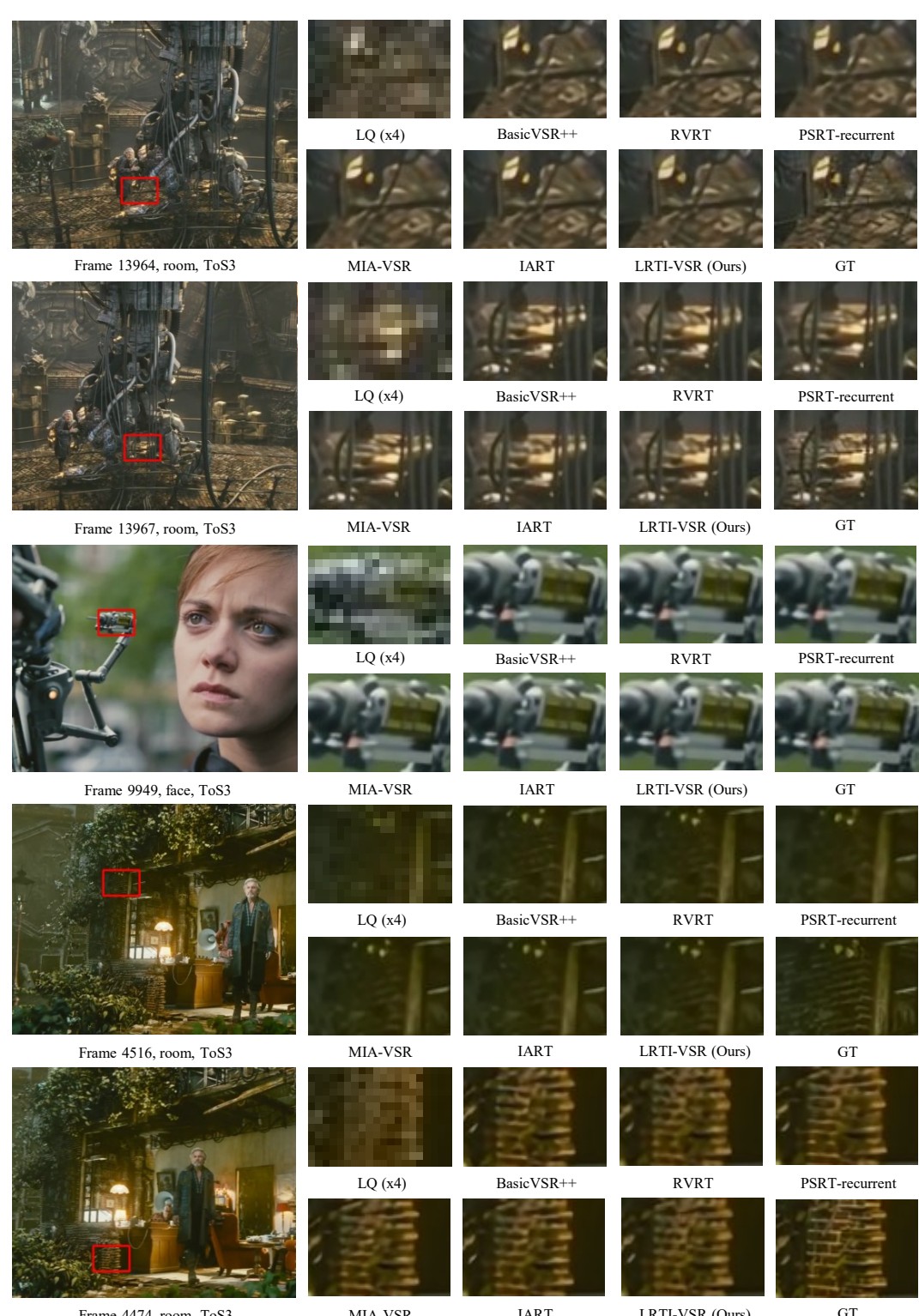

Figure 11: Visual comparison for $4\times$ VSR on ToS3 dataset.

