# OpenReview forum: "LRTI-VSR: Learning Long-Range Refocused Temporal Information from Short Video Clips"
_ICLR.cc/2026/Conference — ICLR 2026 Conference Withdrawn Submission_

### Official Review · Reviewer_Up1K · 2025-10-25

**Soundness:** 2
**Presentation:** 4
**Contribution:** 2
**Rating:** 4
**Confidence:** 4

**Summary:**

This paper proposes LRTI-VSR, a training framework for video super-resolution that combines a truncated backpropagation strategy with a refocused transformer block. The method aims to efficiently learn long-range temporal dependencies while training on short video clips.

**Strengths:**

1. Comprehensive experiments on standard benchmarks.
2. The training strategy shows practical value for efficient training.
3. Good performance compared to existing methods.

**Weaknesses:**

1. Limited technical novelty: bidirectional propagation and truncated training are well-established concepts.
2. The refocused attention mechanism provides incremental improvement rather than fundamental innovation.
3. Missing comparison with the equal-computation baseline of training with long sequences directly

**Questions:**

None

---

### Official Review · Reviewer_zHRv · 2025-10-27

**Soundness:** 3
**Presentation:** 2
**Contribution:** 2
**Rating:** 4
**Confidence:** 4

**Summary:**

In this paper, the authors propose LRTI-VSR, a novel training framework for recurrent-based VSR that aims to efficiently learn long-range temporal dependencies without the prohibitive cost of training on long video clips. To this end, a tailored strategy is proposed to employ temporal propagation features from long video clips while training on shorter ones. In addition,  a refocused intra&inter-frame transformer block is developed to selectively exploit beneficial temporal information via an attention module. Extensive experiments are conducted on several benchmark datasets and the results show the superior performance of the proposed method.

**Strengths:**

- The main idea technically sounds and this paper is easy to follow.
- Extensive experiments show the proposed method produces superior results.

**Weaknesses:**

- If I understand correctly, the proposed training strategy requires the hidden states for the entire long video clip to be stored during the training phase. In Table 4, the memory cost of different truncated length is provided. In addition to this, it would be better to give a theorical analyses of the memory overhead such that readers can be more clearly aware of the cost.

- Following the first comment, in Table 4, why the variant trained with truncated length of 8 has lower memory cost as compared to the baseline trained on 8-frame video clips. It seems more memory overhead should be achieved since hidden state needs to be stored.

- Suppressing less related tokens in the attention layer is widely studied. In this paper, ReLU^2 is introduced to replace Softmax to achieve this. Previous approaches (e.g., Top-k, temperatured softmax) should be included for discussion and comparison to better validate the effectivness of the proposed one.

- The methods included for performance evaluation are mainly published before 2024. More recent SOTA video SR methods (e.g., [c1-c3]) should be included for comparison to better position the proposed method within the video SR area.

[c1] Motion-guided latent diffusion for temporally consistent real-world video super-resolution
[c2] Learning truncated causal history model for video restoration
[c3] Realviformer: Investigating attention for real-world video super-resolution

**Questions:**

See Weaknesses

---

### Official Review · Reviewer_xt8S · 2025-10-28

**Soundness:** 3
**Presentation:** 3
**Contribution:** 3
**Rating:** 6
**Confidence:** 4

**Summary:**

The paper targets a core challenge in video super-resolution (VSR): how to learn long-range temporal dependencies efficiently while still using them effectively at inference. It introduces LRTI-VSR, a training framework for recurrent VSR that decouples sequence length for forward and backward passes. Concretely, the model first runs forward over long clips to obtain accurate hidden states that capture long-term propagation patterns, then performs backpropagation on short clips while conditioning on those cached states, preserving temporal learning without incurring prohibitive training cost. Building on a bidirectional, second-order propagation backbone, the authors also propose a Refocused Intra- and Inter-frame Transformer Block (RITB) that directly attends to both current features and aligned hidden states, replacing standard attention normalization with a sparse, refocusing activation and adding a gated fusion in the feed-forward pathway to emphasize temporally useful signals and suppress misaligned or irrelevant ones. The framework is presented as model-agnostic: it can be plugged into existing CNN or Transformer recurrent VSR architectures. Experiments across long-video benchmarks and ablations aim to show that this strategy improves detail recovery and temporal consistency while maintaining practical training and inference budgets. Overall, the paper positions LRTI-VSR as a principled way to reconcile long-range temporal learning with efficiency, and as a stronger recurrent baseline via its refocused attention design.

**Strengths:**

1.	This paper proposes a truncated training scheme of “long-sequence forward + short-clip backward”. It first performs forward passes on long videos to obtain more accurate hidden states, then backpropagates efficiently on short clips, balancing the learning of long-range dependencies and training efficiency.
2.	The training framework is general for recurrent VSR models and functions as a plug-and-play training paradigm that can be transferred to existing bidirectional and recursive architectures to enhance the representation and utilization of long-range information.
3.	The paper introduces RITB, which applies a sparse refocus activation in the attention stage to suppress irrelevant correlations and strengthen beneficial associations.
4.	A refocused gated unit is introduced in the FFN to fuse the aligned previous-frame hidden state, addressing the selective utilization of cross-frame information.

**Weaknesses:**

1.	RITB replaces SoftMax with ReLU2 to “sparsify/refocus” attention, but it lacks a deeper theoretical comparison regarding stability, gradient properties, and contrasts with SoftMax attention; providing such analysis could help readers gain a more thorough understanding of the proposed method.
2.	Both RITB and RGU rely on the “aligned previous-frame hidden state.” If there are errors in optical flow/deformation estimation or large non-rigid motions, the risk of error propagation and the corresponding suppression mechanisms are not discussed in sufficient detail.

**Questions:**

Please see the weaknesses part.

---

### Official Review · Reviewer_kqu6 · 2025-11-01

**Soundness:** 3
**Presentation:** 2
**Contribution:** 2
**Rating:** 4
**Confidence:** 4

**Summary:**

The paper proposes LRTI-VSR, a recurrent video super-resolution framework that learns long-range temporal dependencies while keeping training costs acceptable. The key idea is to decouple sequence lengths for training: run forward propagation on long sequences to obtain accurate hidden states for all frames, then perform backpropagation on short clips, enabling efficient learning of long-term propagation patterns without modifying base architectures.  The model also introduces a refocused intra- and inter-frame Transformer block that replaces SoftMax with the ReLU² activation in attention to prioritize useful propagated features, and adds a refocused gate that injects aligned hidden states into the FFN to better exploit inter-frame information.  Empirically, LRTI-VSR achieves state-of-the-art or better results on REDS4 and ToS3 at ×4 SR with competitive FLOPs, outperforming strong baselines such as BasicVSR++, RVRT, PSRT-recurrent, MIA-VSR, and IART.

**Strengths:**

- The paper introduces a generic LRTI-VSR training framework that efficiently learns accurate long-term temporal dependencies from long video sequences while maintaining manageable training overhead.

- It proposes a truncated backpropagation strategy that runs forward propagation on long video sequences to obtain temporal propagation hidden states H for all frames, then performs backpropagation on short video clips, enabling the model to learn temporal propagation patterns with high training efficiency.

- The method designs a Refocused intra&inter Transformer Block (RITB) that replaces SoftMax with the sparse refocus activation ReLU² in attention and adds a refocused gate to inject aligned hidden states into the FFN, selectively prioritizing useful inter-frame information and suppressing irrelevant features.

**Weaknesses:**

- From Table 5 and the ablations, most gains stem from the truncated backpropagation training design rather than architectural novelty; the strategy consistently raises PSNR across baselines with minimal parameter change, suggesting optimization dominates architecture-driven progress.

- On the architecture side, improvements seem mainly driven by introducing ReLU² in attention; this reads as a stronger baseline tweak rather than a substantive innovation, as the activation swap alone may not justify a distinct methodological contribution.

- Typesetting needs refinement: tables should be repositioned for narrative flow (e.g., move Table 1 to the Experiments section). Additionally, Figure 2 contains a typographical error—“paopagation” should read “propagation.”

**Questions:**

- Could the proposed training framework be extended to other video restoration tasks—interpolation, deblurring, denoising—by decoupling long-sequence forward passes from short-clip backpropagation? The idea is compelling; demonstrating broader applicability, even without architectural innovations, would significantly strengthen the paper by evidencing generality, and revealing task-agnostic benefits.

---

### Note · Authors · 2025-11-14

I have read and agree with the venue's withdrawal policy on behalf of myself and my co-authors.